# Leading causes of death and high mortality rates in an HIV endemic setting (Kisumu county, Kenya, 2019)

**Anthony Waruru**[1][*], **Dickens Onyango**[2,3,4], **Lilly Nyagah**[5‡], **Alex Sila**[6‡], **Wanjiru Waruiru**[6‡], **Solomon Sava**[7‡], **Elizabeth Oele**[2‡], **Emmanuel Nyakeriga**[6‡], **Sheru W. Muuo**[6], **Jacqueline Kiboye**[8], **Paul K. Musingila**[1], **Marianne A. B. van der Sande**[3,4], **Thaddeus Massawa**[9‡], **Emily A. Rogena**[10], **Kevin M. DeCock**[1], **Peter W. Young**[1]

**1** Division of Global HIV and TB, US Centers for Disease Control and Prevention, Nairobi, Kenya, **2** Kisumu County Department of Health, Kisumu, Kenya, **3** Department of Public Health, Institute of Tropical Medicine, Antwerp, Belgium, **4** Julius Global Health, Julius Centre for Health Sciences and Primary Care, University Medical Centre, Utrecht, Netherlands, **5** Ministry of Health, National AIDS and STI Control Programme, Nairobi, Kenya, **6** Global Programs for Research and Training, Nairobi, Kenya, **7** Jaramogi Oginga Odinga Teaching and Referral Hospital, Kisumu, Kenya, **8** Civil Registration Services, Kisumu County, Kenya, **9** Kisumu County Referral Hospital, Kisumu, Kenya, **10** Department of Human Pathology, School of Medicine, College of Health Sciences, Jomo Kenyatta University of Agriculture and Technology, Nairobi, Kenya

☯ These authors contributed equally to this work.
‡ LN, AS, WW, SS, EO, EN and TM also contributed equally to this work.
* awaruru@cdc.gov

**Data Availability Statement:** Data are available in figshare using the link https://figshare.com/s/e14b72ff064d7bc5467a.

**Funding:** Attribution of support: This publication was made possible by support from the U.S.

## Abstract

### Background

In resource-limited settings, underlying causes of death (UCOD) often are not ascertained systematically, leading to unreliable mortality statistics. We reviewed medical charts to establish UCOD for decedents at two high volume mortuaries in Kisumu County, Kenya, and compared ascertained UCOD to those notified to the civil registry.

### Methods

Medical experts trained in COD certification examined medical charts and ascertained causes of death for 456 decedents admitted to the mortuaries from April 16 through July 12, 2019. Decedents with unknown HIV status or who had tested HIV-negative >90 days before the date of death were tested for HIV. We calculated annualized all-cause and cause-specific mortality rates grouped according to global burden of disease (GBD) categories and separately for deaths due to HIV/AIDS and expressed estimated deaths per 100,000 population. We compared notified to ascertained UCOD using Cohen's Kappa (κ) and assessed for the independence of proportions using Pearson's chi-squared test.

### Findings

The four leading UCOD were HIV/AIDS (102/442 [23.1%]), hypertensive disease (41/442 [9.3%]), other cardiovascular diseases (23/442 [5.2%]), and cancer (20/442 [4.5%]). The all-

President's Emergency Plan for AIDS Relief (PEPFAR) through cooperative agreements U2GGH001520-01 to University of California San Francisco (WW) and GH001953 to the the National AIDS and STI Control Programme (LN) from the U. S. Centers for Disease Control and Prevention (CDC), Division of Global HIV &TB (DGHT). Disclaimer: The findings and conclusions in this report are those of the authors and do not necessarily represent the official position of the funding agencies.

**Competing interests:** The authors have declared that no competing interests exist.

**Abbreviations:** COD, causes of death; CRVS, civil registration, and vital statistics; DOA, dead on arrival; GBD, global burden of diseases; HIV/AIDS, human immunodeficiency virus/acquired immunodeficiency syndrome; ICD10, international classification of diseases and health conditions version 10; TB, tuberculosis; UCOD, the underlying cause of death.

cause mortality rate was 1,086/100,000 population. The highest cause-specific mortality was in GBD category II (noncommunicable diseases; 516/100,000), followed by GBD I (communicable, perinatal, maternal, and nutritional; 513/100,000), and III (injuries; 56/100,000). The HIV/AIDS mortality rate was 251/100,000 population. The proportion of deaths due to GBD II causes was higher among females (51.9%) than male decedents (42.1%; p = 0.039). Conversely, more men/boys (8.6%) than women/girls (2.1%) died of GBD III causes (p = 0.002). Most of the records with available recorded and ascertained UCOD (n = 236), 167 (70.8%) had incorrectly recorded UCOD, and agreement between notified and ascertained UCOD was poor (29.2%; κ = 0.26).

## Conclusions

Mortality from infectious diseases, especially HIV/AIDS, is high in Kisumu County, but there is a shift toward higher mortality from noncommunicable diseases, possibly reflecting an epidemiologic transition and improving HIV outcomes. The epidemiologic transition suggests the need for increased focus on controlling noncommunicable conditions despite the high communicable disease burden. The weak agreement between notified and ascertained UCOD could lead to substantial inaccuracies in mortality statistics, which wholly depend on death notifications.

## Introduction

The World Health Organization (WHO) classifies health problems into three broad categories: Group I includes communicable diseases (including HIV/AIDS) as well as perinatal, maternal, and nutritional diseases; Group II includes noncommunicable diseases; and Group III includes injuries [1, 2]. These classifications are used to calculate the global burden of disease (GBD) causes of death (COD) reports and to provide broad groupings of COD for comparing mortality rates between countries [3].

The overall crude mortality rate in Kenya is estimated to be 550/100,000 population [4]. Estimated all-cause mortality rates increased from 850/100,000 in 1990 to 902/100,000 in 2006 but subsequently decreased to 519/100,000 population by 2016 [5]. Infectious diseases were the biggest contributor to mortality rates before the mid-2000s, driven largely by the impact of HIV. However, from 2006 to 2016, deaths due to infectious diseases, especially HIV/AIDS, have decreased; noncommunicable diseases and injuries account for an increasing fraction of deaths [5–7]. Among noncommunicable diseases, cancer is a major underlying cause of death (UCOD) even in rural settings [6], and approximately 100,000 Kenyans die of hypertension-related complications every year [8]. Hypertension further contributes to 50% of hospital admissions and over 40% of deaths in Kenya [9]. This epidemiological transition may be due to changes in population dynamics and other individual and environmental factors. The leading recertified UCOD in Kenya (estimated in 2016) were; HIV (11.0%), lower respiratory infections (9.1%), malaria (5.7%), non-HIV related tuberculosis (4.0%), diarrhoeal diseases (3.9%), prematurity and low birth weight (3.7%), digestive diseases (3.5%) and anemia (3.3%) [9]. Pneumonia, malaria, and cancer were leading COD in 2017 and were leading causes of morbidity. In 2017, deaths due to HIV/AIDS ranked fifth (8,800) but had declined by 55% between 2010 and 2018 [10]. In 2017, of other communicable diseases, deaths due to tuberculosis ranked fourth; among deaths due to injuries, road traffic accidents were the ninth overall

leading COD [11]. The number of deaths due to road-traffic accidents is estimated to have increased by 8% from 2,907 in 2014 to 3,153 in 2018 [12].

In Kenya, mortality rates for children aged <5 years were 46.37 deaths per 1000 live births in 2018, a gradual decrease from 164.34 deaths per 1000 live births in 1969 [13]. However, 26/47 (55.3%) counties have not met the World Summit for Children target to reduce mortality rates to 70 deaths per 1000 live births by the year 2000. Only nine counties were on course to meet the millennium development goal to reduce mortality rates in children aged <5 years by two-thirds between 1990 and 2015 [13]. In 2009, the adult mortality rate (probability of dying between ages 15 and 60 years per 1000 population) was 348 among men and 313 among women [14].

Although summarizing mortality rates using GBD classes is important, accurate and specific UCOD data are needed to guide and evaluate appropriate public health responses in preventive and curative services. The accuracy of mortality statistics can be improved by ascertaining the most probable COD and ensuring that death records include the correct UCOD. However, in resource-limited countries, UCODs recorded in civil registration and vital statistics (CRVS) systems are commonly determined by individual administrators or health providers, depending on the place of death, rather than on a systematic review of evidence in medical charts or autopsy. Often this is due to lack of medical attention at the time of death, a situation that could be remedied by a systematic assembly of health records, among other approaches [15]. Thus, to improve vital statistics, post-mortem examinations complemented by a hospital-based review of clinical charts can help determine the UCOD. Methodologies for mortuary-based surveillance do not have to be invasive or logistically challenging. For example, minimally invasive autopsy techniques, such as oral swabs, have been used to improve vital statistics in outbreak investigations for diseases such as Ebola [16]. Such methods have utility in public health surveillance, and when combined with available medical history data, the most likely UCOD can be deduced. We ascertained UCOD, antecedent COD, and immediate COD for hospital-based deaths that occurred in two high-volume referral hospitals in Kisumu County, Kenya, compared them to those notified to the civil registry, and estimated mortality rates.

## Methods

### Setting

Kisumu County had an estimated population of 1,155,574 in 2019 [17], and has a high HIV burden [18]. The two largest mortuaries in the county are located at Jaramogi Oginga Odinga Teaching and Referral Hospital (JOOTRH) and Kisumu County Referral Hospital (KCRH), with the capacity to hold 99 and 46 bodies, respectively. There are three civil registries in the county (Kisumu East, Kisumu West, and Nyando). Kisumu East civil registry receives death notifications from Kisumu city and the surrounding areas. In 2019, notifications from the two mortuary facilities accounted for 42.0% of all reported deaths registered in the Kisumu East civil registry [19]. Deaths in Kisumu East civil registry for 3 years (2017–2019) contributed to a median of 74.8% of deaths registered in the entire county [20]. Cause of death data from the two hospitals were used to estimate the number of deaths at the county level with the estimated county population as a denominator.

### Study design and population

This study was part of a more extensive cross-sectional surveillance study to understand HIV-associated mortality in Kisumu County through determining prior HIV diagnosis, HIV-positivity, and viral load among cadavers; assessing the feasibility of using oral fluid obtained from

cadavers for non-invasive rapid HIV antibody testing; and assessing the quality of the UCOD certification, HIV status documentation, and efficiency of death notification in Kisumu County.

## Inclusion and exclusion criteria

Three categories of decedents from the two mortuaries were included in the larger study: hospital deaths (all deaths occurred in the hospital wards or outpatient department), dead on arrival (DOA; deaths occurred elsewhere either outside of hospitals or in other hospitals with subsequent transfer), and police cases (DOA cases or hospital deaths that required a post-mortem examination for legal reasons). Data for this manuscript were drawn from the larger study to establish the leading COD. They included all available medical files of the cadavers admitted to the two morgues during April 16–July 12, 2019, for hospital-based deaths of patients of any age. This analysis included decedents who died within the two hospitals whose medical records were available and excluded all decedents that were DOA, as they did not have medical history records at the hospitals. Decedents with unavailable death notification forms and medical charts were excluded.

## Sample size

The sample size of 690 for the larger study was powered to measure HIV prevalence of 5% among deaths of individuals aged ≥15 years and was adjusted to account for the loss of specimens and ineligibility. All children aged <15 years who died and were admitted to one of the two mortuaries during the study period were also included to cover all decedents admitted to the mortuaries during the study period.

## Ascertaining COD

For the purposes of our study, a panel of six medical officers and two health records information officers were trained by master certification and coding trainers on International Classification of Diseases and Health Conditions, version 10 (ICD10) certification, and coding rules [21]. Afterward, the medical officers used a data entry form modified from the standard death notification form to abstract clinical information for hospital-based deaths from medical records. The following details were abstracted: signs and symptoms of illnesses preceding death, clinical diagnoses, and results of investigations, including HIV status and HIV treatment status. Individual panel members used the abstracted information to assign immediate COD, antecedent COD, and UCOD, and recorded them in a tool. Whenever UCOD was unclear, a panel discussion was held to determine and assign the most probable UCOD. The health records information officers then assigned the actual ICD10 code. These data were directly entered into an open data kit tool with logic checks for accuracy and consistency and submitted to a central database.

## Routinely notified versus panel-assigned COD

We abstracted immediate COD, antecedent COD, and UCOD as documented from notification forms for all deaths that occurred in the two hospitals. In Kenya, death notification forms are divided into two parts and are completed by trained medical officers in triplicate. The first part contains the burial permit, and the second part records the UCOD, antecedent COD, immediate COD, and other significant COD. The second part is submitted to the CRVS department for further analysis, reporting, and archiving in the permanent vital statistics within Kenya. In our study, a panel of trained medical experts revised the COD using the

procedures described. As with the UCOD assigned by the medical expert panel, the deaths documented in the death notification forms were entered into the open data kit tool and were submitted to a central database. HIV status was ascertained during the study, and UCOD assigned and coded by the medical experts' panel, were used for surveillance purposes, and were not used to update CRVS information or decedents' medical records.

## Outcomes

For decedents with unavailable HIV status, HIV status was ascertained using post-mortem testing in the larger study, as described elsewhere [22]. HIV-associated mortality was defined as any death with a documented HIV-positive status, either in the medical files or post-mortem testing results. HIV/AIDS cause-specific mortality was considered for all decedents who had HIV assigned as their UCOD (ICD-10 codes B20-B24), per ICD-10 guidelines. Other COD were as assigned based on ICD-10 coding rules.

**Summary measures.** Two parameters were used to calculate the summary rates. First, we considered the contribution of the two mortuaries to deaths in Kisumu East civil registry (42.0% in 2017) [19]. We then considered the contribution of the Kisumu East civil registry to deaths in the entire county for 2017–2019 (median, 74.8%). Based on this coverage, we calculated the crude all-cause mortality rate for Kisumu County as the total number of deaths attributed to all causes reported during the study period (n = 938), projected to 100% and annualized for 12 months. Rates were expressed per 100,000 of the mid-year population of Kisumu County [17], as shown in the formula:

$$CMR = \left(\frac{d}{p}\right) \times 100,000$$

Where $d$ is the annualized deaths in Kisumu County (i.e., deaths reported during the study period calculated for 12 months), and $p$ is the mid-year population for Kisumu County.

To calculate the cause-specific mortality rate, we used the proportion of deaths attributable to specific causes reported during the study period multiplied by the estimated deaths attributed to specific causes based on the annualized number of deaths. These rates were projected to 100% for the entire county and finally expressed per 100,000 population as shown in the formula:

$$CSMR = c \times \left(\frac{d}{p}\right) \times 100,000$$

Where $c$ is the proportion of deaths reported during the study period attributed to a specific cause; $d$ is the annualized deaths in Kisumu County (i.e., deaths reported during the study period calculated for 12 months); and $p$ is the mid-year population for Kisumu County.

We summarized age-specific mortality rates by sex per 100,000 population and plotted these log-transformed rates for graphical interpretation.

**Stillbirths.** Stillbirths were not subjected to UCOD ascertainment by the panel, but we documented and quantified the occurrence of stillbirths at the two hospitals during the study period. We used the Kenya health information system data to determine the number of deliveries in the two facilities during April 1–July 31, 2019, and calculated the stillbirth rate per 1,000 deliveries.

## Statistical analysis

COD were summarized using the free Microsoft Excel-based tool *Analysing Mortality Levels and Causes-of-Death* (ANACoD V2.0), developed by WHO in collaboration with the

University of Queensland and Health Metrics Network (https://www.who.int/healthinfo/anacod/en/) [23]. This tool provides a stepwise approach to a comprehensive analysis of ICD-10–coded data. We reviewed coded mortality data for errors and tabulated and presented the UCOD by age and sex in tables and charts using the tool. The tool also classifies UCOD using the GBD categorization and compares findings with those from other countries. To measure the interrater agreement of notified compared to ascertained COD, we used Cohen's kappa (κ) statistic. We used the Pearson chi-squared test to compare proportions, and when the n-values were <5, we used the Fisher exact test. P-values <0.05 were considered statistically significant.

### Ethical considerations

No individual personally identifiable information was included in the study database. Results of HIV tests conducted on decedents were not linked back to living individuals. This study was approved by the Kenya Medical Research Institute's Science and Ethical Review Unit; the JOOTRH ethics review committee; U.S. Centers for Disease Control and Prevention (CDC), Center for Global Health, Associate Director for Science, as research involving data and specimens from deceased persons; and University of California San Francisco Committee on Human Research.

## Results

### Overview of included deaths

Of the 938 deaths enrolled in the larger study, 851 (90.7%) cadavers were eligible, and 555/851 (65.2%) died within the two hospitals. The rest (296/851; 35.8%) were either police cases or other DOA cadavers. We retrieved available medical charts and ascertained the probable UCOD for 456 decedents, representing 82.2% of the hospital-based deaths during the study period. The UCOD was not confirmed for 99/555 (17.8%) of the cadavers due to missing hospital records. Of the decedents with available records (n = 456), 442 had UCOD that were classified, and 14/456 (3.1%) deaths could not be classified due to ill-defined diseases (ICD10 codes R00-R99; Fig 1). Therefore, 52% of decedents admitted to the mortuaries (442/851) during the study period could be included in our secondary analysis.

Of the 442 decedents whose UCOD were ascertained and classified, 51 (11.6%) were children aged <1 year, and 42 (9.5%) were aged >80 years. Children aged 10–14 years and young people aged 20–24 years accounted for the lowest proportions of decedents whose UCOD were ascertained and classified (2.7% each). Overall, a similar number of female (234) and male decedents (208) had their UCOD established and classified (Table 1).

The prevalence of GBD group I causes was similar among men (49.0%) and women (45.7%; p = 0.486). Significantly more women (52.1%) than men (42.3%) died due to GBD group II causes (p = 0.039), and significantly more men (8.7%) than women (2.1%) died due to GBD group III causes (p = 0.002).

The estimated age-specific number of deaths pyramid was inverted compared to the population pyramid from the 2019 census (Fig 2). The largest age band for the population is 10–14 years. Children that were aged <5 years and adults aged ≥70 years had the highest numbers of deaths/100,000 population. Higher mortality rates were reported for male decedents aged <1 year, 15–19 years, and 55–59 years, whereas female decedents aged 30–34 years, 45–49 years, and ≥80 years had higher mortality rates.

### Twenty leading causes of death

The 20 leading COD accounted for over three-quarters (77.1%, 341/442) of the ascertained deaths (Table 2). The four leading UCOD were HIV/AIDS (102/442 [23.1%]), hypertensive

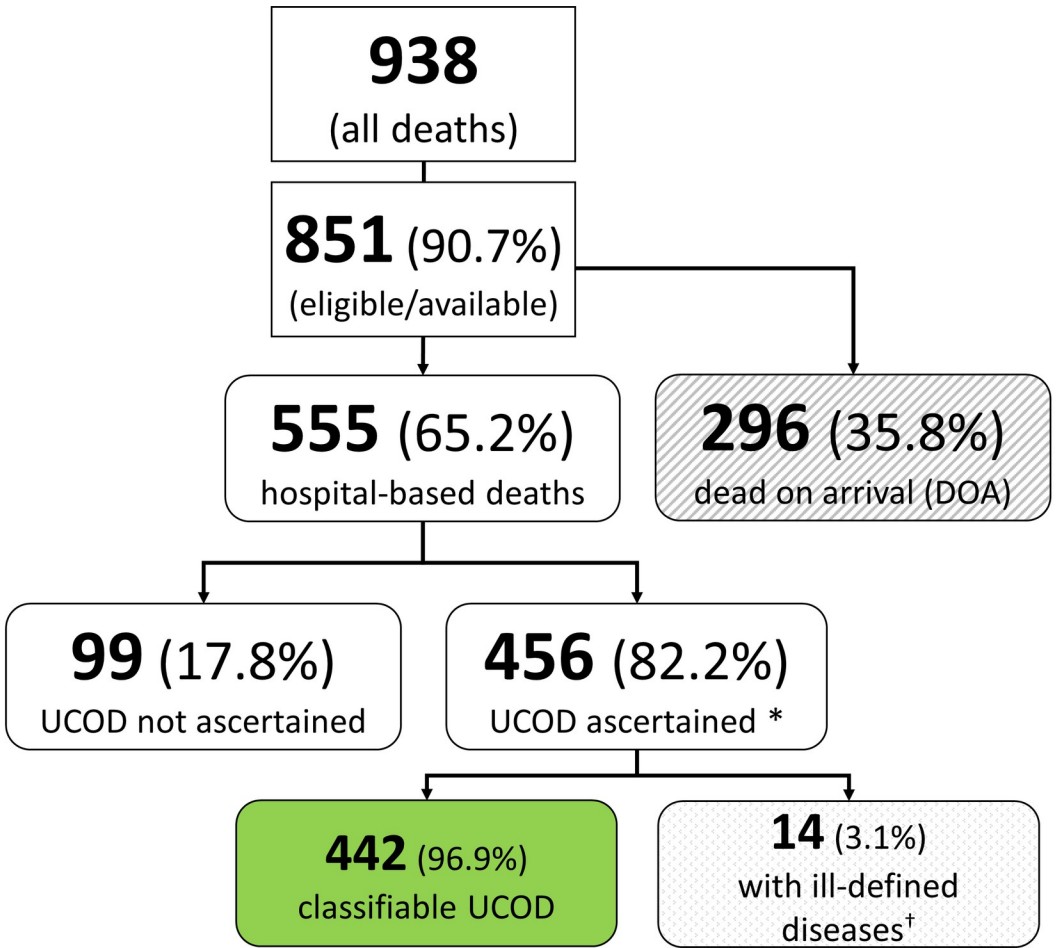

*Hospital records were available. †Ill-defined diseases refer to UCOD with ICD10 codes R00-R99.

**Fig 1. Deaths of hospitalized patients at two referral hospitals, Kisumu County, Kenya (2019).** The figure presents the data flow for the study and the analysis for this manuscript.

disease (41/442 [9.3%]), other cardiovascular diseases (23/442 [5.2%]), and cancer (20/442 [4.5%]). Lower respiratory infections were important UCOD among male decedents (5.3% of deaths) but were ranked twelfth among female decedents (2.1% of deaths). More men had UCOD due to road traffic accidents, ranking sixth, but only three traffic accident-related deaths were reported among women. Significantly more women (12.8%) than men (5.3%) died of hypertensive disease (p = 0.006) and cancer (women, 6.8%; men, 1.9%; p = 0.013).

Among children aged <5 years, conditions arising during the perinatal period (n = 12 [18.8%]), followed by prematurity and low birth weight (n = 11 [17.2%]), were the first and second leading COD among 64 children. HIV/AIDS was the third leading COD among children aged <5 years (Table 3).

## Stillbirths

In both hospitals, there were 66 documented stillbirths during the study period. From health services program data reported through the Kenya health information system, reported 3,360 total deliveries in the two facilities during April 1–July 31, 2019, and 81

**Table 1. Distribution of ascertained causes of death by age and sex, and global burden of disease category from admissions at two large hospitals, Kisumu County, Kenya (2019).**

|  |  | Group I[†] | | Group II[‡] | | Group III[§] | |
|---|---|---|---|---|---|---|---|
|  | **All** | **Male** | **Female** | **Male** | **Female** | **Male** | **Female** |
| **Age (years)** | **N (%*)** | **n (%)** | **n (%)** | **n (%)** | **n (%)** | **n (%)** | **n (%)** |
| 0 | 51(11.5) | 26 (25.5) | 19 (17.8) | 3 (3.4) | 3 (2.5) | 0 (0) | 0 (0) |
| 1–4 | 16(3.6) | 8 (7.8) | 5 (4.7) | 1 (1.1) | 2 (1.6) | 0 (0) | 0 (0) |
| 5–9 | 20(4.5) | 5 (4.9) | 5 (4.7) | 4 (4.5) | 6 (4.9) | 0 (0) | 0 (0) |
| 10–14 | 12(2.7) | 3 (2.9) | 4 (3.7) | 2 (2.3) | 3 (2.5) | 0 (0) | 0 (0) |
| 15–19 | 19(4.3) | 5 (4.9) | 2 (1.9) | 6 (6.8) | 4 (3.3) | 2 (11.1) | 0 (0) |
| 20–24 | 12(2.7) | 3 (2.9) | 3 (2.8) | 2 (2.3) | 3 (2.5) | 1 (5.6) | 0 (0) |
| 25–29 | 37(8.4) | 2 (2) | 13 (12.1) | 10 (11.4) | 5 (4.1) | 6 (33.3) | 1 (20) |
| 30–34 | 41(9.3) | 10 (9.8) | 17 (15.9) | 6 (6.8) | 8 (6.6) | 0 (0) | 0 (0) |
| 35–39 | 26(5.9) | 9 (8.8) | 11 (10.3) | 0 (0) | 3 (2.5) | 3 (16.7) | 0 (0) |
| 40–44 | 24(5.4) | 6 (5.9) | 7 (6.5) | 7 (8) | 3 (2.5) | 1 (5.6) | 0 (0) |
| 45–49 | 23(5.2) | 3 (2.9) | 6 (5.6) | 5 (5.7) | 8 (6.6) | 1 (5.6) | 0 (0) |
| 50–54 | 18(4.1) | 3 (2.9) | 2 (1.9) | 3 (3.4) | 8 (6.6) | 2 (11.1) | 0 (0) |
| 55–59 | 18(4.1) | 4 (3.9) | 2 (1.9) | 6 (6.8) | 6 (4.9) | 0 (0) | 0 (0) |
| 60–64 | 21(4.8) | 5 (4.9) | 4 (3.7) | 4 (4.5) | 6 (4.9) | 1 (5.6) | 1 (20) |
| 65–69 | 14(3.2) | 1 (1) | 1 (0.9) | 5 (5.7) | 7 (5.7) | 0 (0) | 0 (0) |
| 70–74 | 29(6.6) | 1 (1) | 3 (2.8) | 11 (12.5) | 13 (10.7) | 1 (5.6) | 0 (0) |
| 75–79 | 19(4.3) | 4 (3.9) | 1 (0.9) | 4 (4.5) | 9 (7.4) | 0 (0) | 1 (20) |
| 80+ | 42(9.5) | 4 (3.9) | 2 (1.9) | 9 (10.2) | 25 (20.5) | 0 (0) | 2 (40) |
| **Total** | **442** | **102** | **107** | **88** | **122** | **18** | **5** |
| p-values[¶] | - | 0.486 | | 0.039 | | 0.002 | |

*Column percentage.

[†]Group I–Communicable, perinatal, maternal, and nutritional including HIV.

[‡]Group II–Noncommunicable diseases.

[§]Group III–Injuries.

[¶]P-values calculated using chi-squared test show significance of differences in proportions for each GBD category by sex.

(2.4%) stillbirths over 121 days [20]. Using this denominator, we calculated approximately 2,416 deliveries in the 87 days of the study period and a stillbirth rate of 2.7% (66/2,416) or 27/1,000 deliveries.

## Comparison of notified versus ascertained underlying causes of death

We abstracted data from available death notification forms for 236 records that were matched with UCOD records ascertained by the expert panel. Overall, over two-thirds (167/236 [70.8%]) of the decedents had incorrectly assigned UCOD in the death notification form (Table 4). The errors were attributed to either wrong sequencing, i.e., the chain of events leading to death were incorrectly ordered 10/167 [6.0%]), or incorrect assignment (157/167 [94.0%]) of UCOD among decedents. The proportion of incorrectly assigned UCOD was not significantly different by sex (p = 0.174). Where underlying UCOD was HIV/AIDS, the concordance between panel-ascertained and notified deaths was higher (27/60 [45.0%]) compared to other UCOD (42/176 [23.9%]; p = 0.002). Agreement between notified and ascertained UCOD was 29.2% (κ = 0.259). The discrepancy was higher for immediate COD (176/236 [74.6%]) than for UCOD (167/236 [70.8%]).

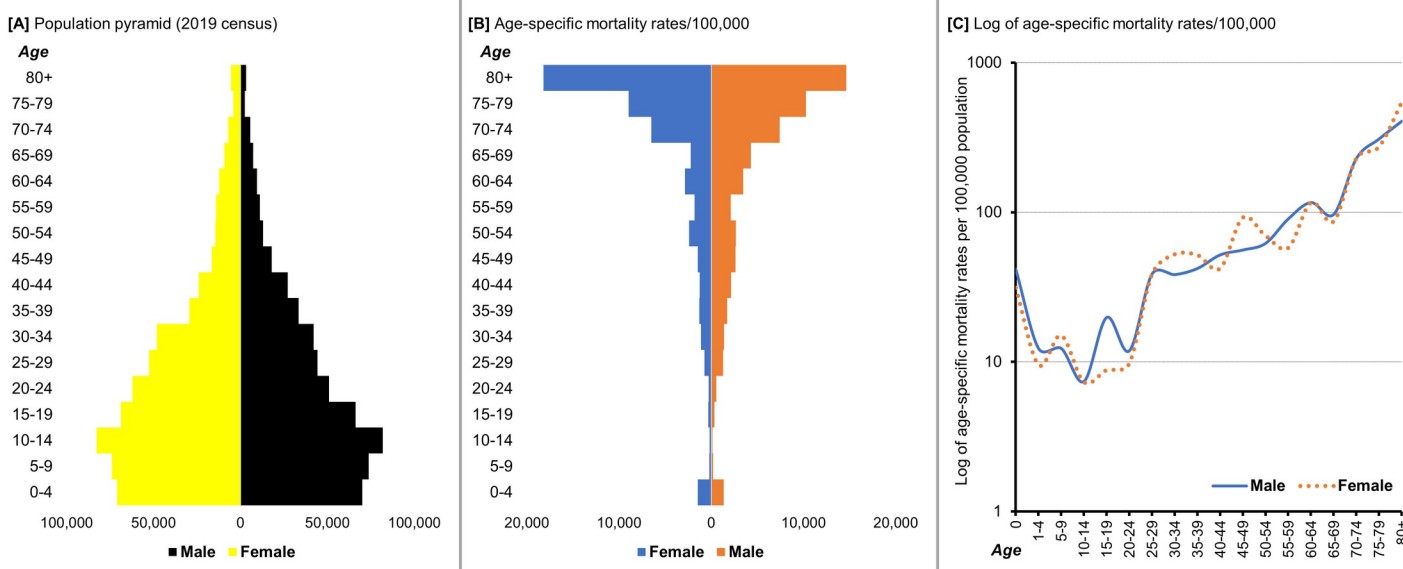

**Fig 2. Population pyramid and estimated deaths per 100,000 by sex and age, Kisumu County, Kenya (2019).** A) Source of population data is 2019 census, B) Estimated deaths/100,000 calculated using the population denominator, C) Log-transformed age-specific mortality rates/100,000 population.

### Summary mortality rates

The all-cause mortality rate among all decedents was 1,086/100,000 population. Noncommunicable diseases contributed to the highest cause-specific mortality (516/100,000 population), followed by GBD I (513/100,000 population) and III (56/100,000 population). Men (81,154/100,000 population) had a higher crude all-cause mortality rate compared to women (1,021/100,000 population). Among decedents aged <15 years at death, the crude all-cause mortality rate was 524/100,000 population, and most deaths were attributed to communicable diseases (397/100,000 population compared to noncommunicable diseases [127/100,000 population]). Among decedents aged ≥15 years at death, men had the highest rate (1,594/100,000 population; women, 1,315/100,000 population), and GBD category II diseases were the leading COD (785/100,000 population compared to GBD I [566/100,000 population] and GBD III [97/100,000 population]). The HIV-associated mortality rate was 359/100,000 population among all decedents. Women and girls of any age (274/100,000 population) had a slightly >20% higher mortality rate associated with HIV than boys and men (224/100,000 population), and deaths that were directly attributed to HIV/AIDS were nearly 40% higher among girls and women (413/100,000 population) than among boys and men (298/100,000 population). The rate of death due to HIV/AIDS was much greater among persons aged ≥15 years (388/100,000 population) compared to younger individuals (53/100,000 population) (Table 5).

## Discussion

### Leading COD

In Kisumu County, we found that the HIV/AIDS-related mortality rate is nearly 25%, which is similar to that observed in Abidjan by De Cock et al. during the late 1980's [24] and double the rate found in a Nairobi mortality study (12.6%) [25]. In our study, the proportion of deaths attributed to HIV/AIDS was twice the proportion that is regularly reported through CRVS. However, the proportion is likely overestimated in our study because we did not include the DOA group. The prevalence of HIV infection was 23.7% among DOA compared to 31.0%

**Table 2. Ascertained leading causes of death at two large hospitals and among all persons and sex, Kisumu County, Kenya (2019).**

| Causes of death | Total (n = 442) | UCOD by sex, rank, n (%)* | | |
| --- | --- | --- | --- | --- |
| | Rank, n (%) | Male | Female | p-value[†] |
| *Leading causes* | *n = 341 (77.1)* | *n = 149 (71.6)* | *n = 192 (82.1)* | - |
| HIV/AIDS | **1**,102 (23.1) | **1**, 43 (20.7) | **1**, 59 (25.2) | 0.258 |
| Hypertensive disease | **2**, 41 (9.3) | **2**, 11 (5.3) | **2**, 30 (12.8) | 0.006 |
| Other cardiovascular diseases | **3**, 23 (5.2) | **4**, 10 (4.8) | **4**, 13 (5.6) | 0.724 |
| Cancer[‡] | **4**, 20 (4.5) | **11**, 4 (1.9) | **3**, 16 (6.8) | 0.013 |
| Endocrine disorders | **5**, 18 (4.1) | **5**, 9 (4.3) | **5**, 9 (3.8) | 0.799 |
| Lower respiratory infections | **6**, 16 (3.6) | **2**, 11 (5.3) | **12**, 5 (2.1) | 0.077 |
| Perinatal conditions | **7**, 16 (3.6) | **6**, 8 (3.8) | **6**, 8 (3.4) | 0.810 |
| Other digestive diseases | **8**, 13 (2.9) | **9**, 6 (2.9) | **8**, 7 (3) | 0.947 |
| Malaria | **9**, 12 (2.7) | **8**, 7 (3.4) | **11**, 5 (2.1) | 0.428 |
| Cerebrovascular disease | **10**, 12 (2.7) | **11**, 5 (2.4) | **7**, 7 (3) | 0.704 |
| Prematurity & low birth weight | **11**, 11 (2.5) | **9**, 6 (2.9) | **13**, 5 (2.1) | 0.614 |
| Road traffic accidents | **12**, 11 (2.5) | **6**, 8 (3.8) | -[§],3 | 0.125 |
| Diabetes mellitus | **13**, 10 (2.3) | **16**, 4 (1.9) | **10**, 6 (2.6) | 0.755 |
| Diarrheal diseases | **14**, 9 (2) | **11**, 5 (2.4) | **15**, 4 (1.7) | 0.740 |
| Infectious diseases (other) | **15**, 9 (2) | **20**, 3 (1.4) | **9**, 6 (2.6) | 0.510 |
| Birth asphyxia and birth trauma | **16**, 6 (1.4) | **11**, 5 (2.4) | -[§],1 | 0.104 |
| Protein-energy malnutrition | **17**, 6 (1.4) | -[§],3 | **18**, 3 (1.3) | 1.000 |
| Skin diseases | **18**, 6 (1.4) | -[§],1 | **14**, 5 (2.1) | 0.220 |
| *All other causes*[¶] | -, *101 (22.9)* | -, *59 (28.4)* | -, *42 (17.9)* | - |
| **Total (N)** | **442** | **208** | **234** | |

*UCOD: underlying causes of death. Causes of death exclude Ill-defined diseases (ICD10 R00-R99). Three percent of underlying causes of death were/remained ill-defined. Results within columns presented as rank, N (%). Rank is boldfaced.

[†]p-values show significance of differences in proportions for each UCOD by sex

[‡] Cancers include UCOD defined as: "Esophageal cancer", "Cervix uteri cancer" and "Trachea, bronchus and lung cancers", "Other neoplasms", "Other malignant neoplasms",–hence combined cancers reduce the number of leading causes to 18

[§]Not among 20 overall leading UCOD within the sex category, hence percentages and ranking not included

[¶]Includes all other mutually exclusive causes of death, percentages are out of the total number of deaths.

among hospital-based deaths. Among the 47 counties in Kenya, Kisumu County reported the greatest number of HIV/AIDS-related deaths (14.4% of all HIV/AIDS deaths nationally in 2017) [9], compared to only 2.4% of the current national population [17]. Kisumu County is ranked second in adult HIV prevalence at 17.5% [18], and our finding of high HIV-related mortality rates are consistent with the national HIV prevalence. Though our finding of a high proportion of HIV/AIDS-related deaths likely reflects the complexity of cases in the two referral hospitals, our sample did have broad geographic coverage, including decedents from all over the county. However, the difference in our findings compared to CRVS data could be explained by the extra step of ascertaining UCOD in our study; thus, we had higher sensitivity in identifying HIV/AIDS-attributable mortality. We also tested the cadavers for HIV, and the availability of these test results helped us distinguish HIV-related from non-HIV-related deaths with similar clinical presentation.

The proportion of deaths with hypertension as an underlying cause was (9.3%) and greatest among persons aged ≥40 years (36/41; 87.8%) for both men and women, which is below the proportions previously reported of 12.3%, [26], and 22.8%, [27], in informal urban settlements in Kenya. Cancer was the fourth leading COD and accounted for a greater proportion of

**Table 3. Ascertained leading causes of death at two large hospitals and among children aged 0–4 years in Kisumu County, Kenya (2019).**

|  | Causes of death* | n (%)† |
|---|---|---|
| *Rank* | *Leading causes* | *n = 64* |
| **1** | Other conditions arising during the perinatal period | 12(18.8) |
| **2** | Prematurity and low birth weight | 11(17.2) |
| **3** | HIV/AIDS | 6(9.4) |
| **3** | Birth asphyxia and birth trauma | 6(9.4) |
| **4** | Diarrhoeal diseases | 5(7.8) |
| **4** | Protein-energy malnutrition | 5(7.8) |
| **5** | Lower respiratory infections | 4(6.3) |
| **6** | Malaria | 3(4.7) |
| **7** | Lymphomas and multiple myeloma | 2(3.1) |
| **7** | Endocrine disorders | 2(3.1) |
| **8** | Meningitis | 1(1.6) |
| **8** | Other infectious diseases | 1(1.6) |
| **8** | Other nutritional disorders | 1(1.6) |
| **8** | Other malignant neoplasms | 1(1.6) |
| **8** | Other digestive diseases | 1(1.6) |
| **8** | Abdominal wall defect | 1(1.6) |
| **8** | Other Congenital anomalies | 1(1.6) |
| - | *All other causes‡* | 4 |
|  | **Total** | **N = 67** |

*Data for children overlap with data presented in Table 2

†Column percentages

‡Not ranked or included in the percentage.

deaths among women than men. Our findings are consistent with the reports that showed cancer as the third leading COD in Kenya, after infectious diseases and cardiovascular diseases [28], and the second leading COD globally, responsible for an estimated 9.6 million deaths in 2018 [29]. Additionally, our findings point to an epidemiologic transition underway in Kenyan adults in rural settings. This transition highlights the increasing importance of preventing non-communicable diseases and reduce adult mortality rates [6, 7]. Nonetheless, infectious diseases still contribute to significant mortality in this population, and, not surprisingly, most of the deaths among younger decedents were due to infectious diseases [30].

More than a fifth of certified deaths were among infants and among adults aged ≥80 years. Mortality rates among the geriatric population were similar to those previously reported in Kenya (~10%), with similar mortality rates for both men and women [9]. Although high mortality rates are expected among the geriatric population, the high proportion of deaths among infants reflects the persistent high infant mortality rates in the Nyanza region, estimated to be 72 deaths/1,000 live births [31]. Possible contributors to this high infant mortality rate are malaria and child and maternal nutrition risk factors [5]. We also observed a higher proportion of deaths attributed to noncommunicable diseases among women and girls compared to men and boys.

Although exclusion of police cases and DOA likely disproportionately affected our estimates of violence-related deaths (relatively low overall even for men) and injury-related deaths, deaths due to injuries were still four times higher among men than among women, probably due to greater exposure to poor road safety precautions and/or occupational hazards.

**Table 4. Comparison of notified versus the ascertained cause of death (COD) in Kisumu County, Kenya (2019).**

|  | N (%) | Type of errors, n (%) | | p-value |
|---|---|---|---|---|
|  |  | Correct COD | Erroneous COD |  |
| **COD type** |  |  |  |  |
| Underlying | 236 (100) | 69 (29.2)* | 167 (70.8)† |  |
| Immediate | 236 (100) | 60 (25.4)‡ | 176 (74.6) |  |
| **Sex** |  |  |  | 0.174§ |
| Male | 124 (52.5) | 41 (33.1) | 83 (66.9) |  |
| Female | 112 (47.5) | 28 (25.0) | 84 (75.0) |  |
| **Underlying COD** |  |  |  | 0.002§ |
| HIV/AIDS | 60 (25.4) | 27 (45.0) | 33 (55.0) |  |
| Other | 176 (74.6) | 42 (23.9) | 134 (76.1) |  |

*Poor interrater agreement (29.2%; κ = 0.26)

†Wrong assignment of COD (n = 157 [94.0%]) and wrong sequence (n = 10 [6.0%])

‡Poor interrater agreement (25.4%; κ = 0.24)

§p-values show significance of differences in proportions for erroneously assigned COD.

The cause-specific mortality rate for deaths caused by injuries (mostly road traffic-related) was higher for men than for women. Deaths due to road traffic-related injuries could be decreased, especially among younger men, by training passengers, drivers, and other road users on safety precautions.

Surprisingly, we did not find injury-related deaths among children. This is in contrast to studies for other resource-limited countries, which have reported high rates of injury-related deaths among children and younger persons [32, 33]. Injuries are often reported as UCOD for DOA cadavers. Since we did not capture UCOD for DOA cadavers, we probably missed injury-related childhood deaths. However, in our study, hospital-based deaths had a proportionately higher number of decedents aged <5 years at the time of death than those DOA (S1 Fig), whereas more decedents aged <70 years at the time of death were DOA than hospital-based deaths. Thus, the age distribution for DOA compared to hospital-based deaths was similar for most decedents aged

**Table 5. Estimated all-cause and cause-specific mortality rates by GBD and HIV disease classifications in Kisumu County, Kenya (2019).**

|  | Mortality rate per 100,000 population | | | | | | | | |
|---|---|---|---|---|---|---|---|---|---|
| Cause of death | All | | | <15 years old | | | 15+ years old | | |
|  | M/F* | M | F | M/F* | M | F | M/F* | M | F |
| *All-cause* | *1,086* | *1,154* | *1,021* | *524* | *552* | *495* | *1,448* | *1,594* | *1,315* |
| Group I† | 513 | 566 | 467 | 397 | 446 | 348 | 566 | 613 | 521 |
| Group II‡ | 516 | 488 | 532 | 127 | 106 | 148 | 785 | 797 | 760 |
| Group III§ | 56 | 100 | 22 | 0 | 0 | 0 | 97 | 184 | 35 |
| *HIV* |  |  |  |  |  |  |  |  |  |
| HIV-associated¶ | 359 | 298 | 413 | 85 | 81 | 89 | 549 | 455 | 627 |
| Due to HIV/AIDS# | 251 | 224 | 274 | 53 | 40 | 67 | 388 | 362 | 410 |

*Male or female

†Group I–Communicable, perinatal, maternal and nutritional conditions including HIV

‡·Group II–Noncommunicable diseases

§Group III–Injuries

¶HIV was listed as a significant cause of death; #UCOD was ascertained as HIV/AIDS.

5–69 years (S1 Fig). In our study, the proportions of deaths due to communicable diseases compared to noncommunicable diseases were approaching 1:1, similar to those previously reported through CRVS for Kisumu County [19]. However, this ratio is different compared to other similar resource-limited settings in which for every two deaths attributed to communicable diseases, there are three attributed to noncommunicable diseases [34], indicating that HIV still contributes substantially to the burden of communicable diseases in Kisumu. Other infectious diseases contributed to more deaths among children than among adults.

## Stillbirths

We reported a high rate of stillbirths as well as deaths due to perinatal conditions. The high incidence in this setting may be partly due to these referral hospitals attending to complicated pregnancies. These deaths may be related to lack of access to emergency care during pregnancy, late presentation at health care facilities, and complications during childbirth. In Kenya, only about 18% of pregnant women have at least four ANC visits, and women with one visit within the first 3 months of pregnancy have better pregnancy outcomes than those with only one antenatal clinic visit [35]. Teaching pregnant and postpartum women the danger signs of complications during pregnancy and the peripartum period could trigger timely health-seeking behavior [36] and reduce maternal mortality and perinatal deaths [37]. Though our study lasted for only a few months, our estimated stillbirth rate (2.7%) is higher than the national average (1.3%) among women reporting a pregnancy in the 5 years before the 2014 demographic and health survey [31]. Still, it is similar to the 2015 estimate for sub-Saharan Africa (2.9% or 29/1000 live births) [38].

## Annualized mortality rates

We found an all-cause mortality rate of 1,086/100,000 in Kisumu, about twice the national crude death rate average of 551.8/100,000 population for 2015–2020 [4]. Mortality rates have been higher in western Kenya, including Kisumu County [5], which is mostly attributed to communicable diseases, especially HIV/AIDS. However, the rate we found was lower than that reported in a similar setting in neighboring Siaya County (1,446/100,000 population) [39] and is also lower than the 2009 population census estimates of 1,370 deaths/100,000 population [14].

## Notified versus ascertained underlying causes of death

Almost three-quarters of notified deaths at JOOTRH and KCRH had an incorrect UCOD in the death notification forms submitted to the CRVS system. The errors were due to either wrong sequencing or wrong assignment, as noted in previous Ministry of Health reports [9]. The incorrect assignment was higher among male decedents compared to female decedents. Better assignment of underlying UCOD was observed for deaths due to HIV/AIDS, and the proportion with errors was higher for immediate compared to underlying COD. We could not determine whether the higher error rate for immediate COD was due to symptomatic presentation, availability of HIV diagnosis, or treatment records in patient charts. Among persons who died of HIV/AIDS, approximately 50% had one or more noncommunicable diseases documented along the causal pathway as an immediate or intermediate COD. Assignment of the wrong UCOD could be due to various factors. First, the clinician who certified the death may not have seen the patient during hospitalization and thus may not be familiar with the patient's clinical history. Second, medical records may be unavailable or not reviewed by the certifier due to time constraints imposed by the urgency of completing administrative procedures related to the death.

Our study's very low agreement of ascertained UCOD compared to notified UCOD indicates that mortality statistics that depend wholly on death notifications may be grossly inaccurate. In addition, certification of deaths by personnel who are not trained for that purpose may further compromise the quality of mortality data, which is also compounded by poor documentation. This was reflected in a previous study that found a low vital statistics performance index in Kenya [40]. This index is a composite metric that ranges from 0 to 1 for assessing the quality of data on mortality and COD containing six dimensions: quality of COD reporting, quality of reporting of age and sex of decedents, internal consistency, completeness, level of cause-specific detail, and data availability or timeliness [41]. In 2015, Kenya scored 0 in each of these elements [40]. Completeness, correctness, and order of COD statements among decedents have been noted to be poor in Kenya, with incomplete information and citing mechanisms of death as the most frequent errors [42].

Data contained in death notification forms in Kenya are of poor quality, with 18.4% containing ill-defined causes (such as those with incomplete descriptions of the cause or with vague descriptions), which is above the maximum acceptable threshold of 10% for all ages, and about a third of death notification forms were either incomplete or submitted without any COD listed [9]. Certification of deaths by clinicians trained on ICD procedures, availability of HIV status at the time of certification in settings where HIV is a significant COD, and thorough scrutiny of medical records before death certification could substantially improve the quality of UCOD information reported to CRVS.

## Limitations and assumptions

Our study had several limitations. We used available medical records and post-mortem HIV testing to assign COD; no diagnostic autopsies were performed for this study. Group I communicable diseases include perinatal, maternal, and nutrition diseases. However, without post-mortem, there may be no proper differentiation of deaths related to maternal and child health versus communicable diseases for younger decedents. We assumed that both mortality rates and reporting rates were similar in all calendar months for mortality rate estimates. The study was conducted in public health facilities, and we assumed that the cadavers admitted in JOOTRH and KCRH would represent the Kisumu County population. Data abstracted from the Kisumu East Civil Registry indicate that over 75% of deaths occur within health facilities, and about 42% of facility-based deaths are admitted to these two mortuaries. Thus, though facility-based COD statistics may not accurately represent the COD throughout the community, the benefits of estimating COD based on facility-based reported deaths outweigh this bias. To mitigate this limitation, we confirmed that the decedents in our study came from all geographical areas of Kisumu County. Since we did not have medical charts for the DOA (one-third of all decedents during our study period), we assumed the distribution of UCOD for those who were DOA was similar to those who died in the hospitals to calculate mortality rates. These decedents were not selectively admitted to the morgues due to death from any specific GBD category. Regardless, homicides and other violent deaths are underrepresented as COD in our study since we did not include the DOA group in our analysis. This bias may be around 3% of total mortality since the estimated contribution of deaths caused by GBD III category in Kenya is about 8% [43], compared to the 3.8% we found in our analysis.

## Conclusions

Kisumu County had similar communicable and noncommunicable disease related mortality rates, with HIV contributing to the highest proportion of communicable disease-related deaths. Men have higher rates of injury-related deaths than women, whereas women have

higher rates of noncommunicable disease-related deaths. Although most deaths are preventable, our findings suggest that a holistic approach rather than focusing on one GBD category could help prevent deaths in Kisumu County. HIV prevention, improved viral load suppression among HIV-positive persons, and improved uptake of vaccines could help decrease communicable and noncommunicable disease-related deaths. Additionally, prevention and treatment of hypertension and increased screening and community awareness, including advocating for behavior changes such as cessation of smoking, could help decrease noncommunicable disease-related deaths.

Vital statistics reporting for hospital-based deaths could be improved by training medical officers on correct death certification, including proper sequencing using ICD10 rules. For deaths that occur within the community, training the chiefs and sub-chiefs who are responsible for collecting verbal autopsy data could help improve mortality surveillance, as recently reported in Uganda [44]. Routine review of patient charts and data quality reviews by trained medical experts before notification of death could improve the quality of COD reported to CRVS and enhance the utility of such reports for planning. Regular mortality case review meetings may offer a forum to share experiences and ensure that UCOD are documented appropriately.

Our study demonstrates the feasibility of determining probable COD and providing more precise cause-specific mortality rates without using expensive post-mortem procedures. Our approach can be used in other limited-resource settings, particularly regions with high HIV burden, to help evaluate the impact of HIV care and treatment programs and to distinguish deaths caused by HIV/AIDS from other disparate causes and identify epidemiologic transitions.

## Supporting information

**S1 Fig. Age distribution of deaths by type of cadavers admitted to the two morgues at two referral hospitals in Kisumu County, 2019.** The hospital-based deaths had a higher proportion of children aged <5 years compared to dead on arrival (DOA) cadavers. Age distribution was similar for the rest of the cadavers, except for the DOA group, which had proportionately more cadavers aged ≥70 years than the hospital-based group.
(TIF)

## Acknowledgments

We thank the study staff working at the mortuaries and hospitals and the data entry and management staff.

## Author Contributions

**Conceptualization:** Anthony Waruru.

**Data curation:** Anthony Waruru, Alex Sila.

**Formal analysis:** Anthony Waruru.

**Funding acquisition:** Lilly Nyagah, Wanjiru Waruiru.

**Investigation:** Anthony Waruru, Dickens Onyango, Lilly Nyagah, Wanjiru Waruiru, Solomon Sava, Elizabeth Oele, Emmanuel Nyakeriga, Sheru W. Muuo, Thaddeus Massawa, Peter W. Young.

**Methodology:** Anthony Waruru, Lilly Nyagah, Alex Sila, Wanjiru Waruiru, Emmanuel Nyakeriga, Peter W. Young.

**Project administration:** Dickens Onyango, Lilly Nyagah, Wanjiru Waruiru, Emmanuel Nyakeriga, Thaddeus Massawa.

**Resources:** Lilly Nyagah.

**Software:** Alex Sila.

**Supervision:** Anthony Waruru, Dickens Onyango, Lilly Nyagah, Alex Sila, Wanjiru Waruiru, Solomon Sava, Elizabeth Oele, Emmanuel Nyakeriga, Sheru W. Muuo, Jacqueline Kiboye, Paul K. Musingila, Thaddeus Massawa, Peter W. Young.

**Validation:** Lilly Nyagah, Alex Sila, Wanjiru Waruiru, Elizabeth Oele, Paul K. Musingila, Peter W. Young.

**Visualization:** Anthony Waruru.

**Writing – original draft:** Anthony Waruru.

**Writing – review & editing:** Anthony Waruru, Dickens Onyango, Lilly Nyagah, Alex Sila, Wanjiru Waruiru, Solomon Sava, Elizabeth Oele, Emmanuel Nyakeriga, Sheru W. Muuo, Jacqueline Kiboye, Paul K. Musingila, Marianne A. B. van der Sande, Thaddeus Massawa, Emily A. Rogena, Kevin M. DeCock, Peter W. Young.

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
