## [Editor Report · Decision Letter 0]

10 Jun 2021

PONE-D-20-36117

Leading causes of death and high mortality rates in an HIV endemic setting (Kisumu County, Kenya, 2019)

PLOS ONE

Dear Dr. Waruru,

Thank you for submitting your manuscript to PLOS ONE. After careful consideration, we feel that it has merit but does not fully meet PLOS ONE’s publication criteria as it currently stands. Therefore, we invite you to submit a revised version of the manuscript that addresses the points raised during the review process.

This is a well written manuscript. Congratulations on your work! 

Please reformat the paper according to the PLOS One format, and resubmit. (Link: https://journals.plos.org/plosone/s/submission-guidelines)

We look forward to receiving your revised manuscript.

Kind regards,

Sabeena Jalal, MBBS, MSc, MSc, SM

Academic Editor

PLOS ONE

Journal Requirements:

2.  As explained in PLOS ONE's manuscript guidelines (http://journals.plos.org/plosone/s/submission-guidelines#loc-references), we do not allow citation of, or reliance on, unpublished work. Please provide the relevant information in the manuscript and/or a Supporting Information file; if the in reference 20 and 24, is accepted for publication while your PLOS ONE submission is under review, you can remove the Supporting Information file and revert to the citation in your Methods section before acceptance/publication of the current submission.

---

## [Author Response · Author response to Decision Letter 0]

18 Jun 2021

Friday, June 18, 2021

Editors-in-Chief, PLOS ONE

Dear editors,

Thank you for the review of our manuscript, “Leading causes of death and high mortality rates in an HIV endemic setting (Kisumu county, Kenya, 2019)” for publication in PLOS ONE. Thank you also for the positive feedback.

We have considered the editorial comments raised, and we have reviewed the manuscript to reflect the requested changes. We have also modified the text further where necessary to improve on clarity. For example, by changing heading levels where appropriate. While reviewing table 5, we realized an anomaly in the rates for HIV-associated and deaths due to HIV/AIDS (they were swopped for the age dissagerations). We have corrected that anomaly. There are no other changes in the contents of the originally submitted manuscript unless otherwise stated above.

There is no change in the financial disclosure statement. Additionally, we have reviewed the reference list to ensure its completeness and correctness. As a result, we have not retracted any references. References 20 and 24 are published documents.

There are no ethical or legal restrictions on sharing a de-identified dataset. Therefore, we have uploaded the de-identified data file in this location <https://figshare.com/s/e14b72ff064d7bc5467a>, DOI: 10.6084/m9.figshare.14806176.

We hope our paper will receive favorable consideration for publication.

Yours sincerely,

Anthony Waruru, for co-authors

---

## [Decision Letter · Decision Letter 1]

14 Oct 2021

PONE-D-20-36117R1Leading causes of death and high mortality rates in an HIV endemic setting (Kisumu County, Kenya, 2019)PLOS ONE

Dear Dr. Waruru,

Thank you for submitting your manuscript to PLOS ONE. After careful consideration, we feel that it has merit but does not fully meet PLOS ONE’s publication criteria as it currently stands. Therefore, we invite you to submit a revised version of the manuscript that addresses the points raised during the review process. The topic is extremely interesting. However, the paper needs to be re-written. There are grammatical errors in the manuscript. Also, please take note of comments from other editors. Please submit your revised manuscript by Nov 28 2021 11:59PM. If you will need more time than this to complete your revisions, please reply to this message or contact the journal office at plosone@plos.org. Please include the following items when submitting your revised manuscript:A rebuttal letter that responds to each point raised by the academic editor and reviewer(s). You should upload this letter as a separate file labeled 'Response to Reviewers'.A marked-up copy of your manuscript that highlights changes made to the original version. You should upload this as a separate file labeled 'Revised Manuscript with Track Changes'.An unmarked version of your revised paper without tracked changes. You should upload this as a separate file labeled 'Manuscript'.If applicable, we recommend that you deposit your laboratory protocols in protocols.io to enhance the reproducibility of your results. Protocols.io assigns your protocol its own identifier (DOI) so that it can be cited independently in the future. For instructions see: https://journals.plos.org/plosone/s/submission-guidelines#loc-laboratory-protocols. Additionally, PLOS ONE offers an option for publishing peer-reviewed Lab Protocol articles, which describe protocols hosted on protocols.io. Read more information on sharing protocols at https://plos.org/protocols?utm_medium=editorial-email&utm_source=authorletters&utm_campaign=protocols.

We look forward to receiving your revised manuscript.

Kind regards,

Sabeena Jalal, MBBS, MSc, MSc, SM

Academic Editor

PLOS ONE

Journal Requirements:

Reviewers' comments:

Reviewer's Responses to Questions

**Comments to the Author**

1. If the authors have adequately addressed your comments raised in a previous round of review and you feel that this manuscript is now acceptable for publication, you may indicate that here to bypass the “Comments to the Author” section, enter your conflict of interest statement in the “Confidential to Editor” section, and submit your "Accept" recommendation.

Reviewer #1: All comments have been addressed

Reviewer #2: (No Response)

2. Is the manuscript technically sound, and do the data support the conclusions?

Reviewer #1: Yes

Reviewer #2: Yes

3. Has the statistical analysis been performed appropriately and rigorously? 

Reviewer #1: Yes

Reviewer #2: Yes

4. Have the authors made all data underlying the findings in their manuscript fully available?

Reviewer #1: Yes

Reviewer #2: Yes

5. Is the manuscript presented in an intelligible fashion and written in standard English?

Reviewer #1: Yes

Reviewer #2: Yes

6. Review Comments to the Author

Reviewer #1: * statistical analysis was well prepared in this study

- it is interesting to have this conclusion, it gives a good matching between the background and its results.

- it's a well-written article by following a basic scientific English.

* I recommend to be published.

Reviewer #2: Thank you for letting me revise such an interesting paper. It looks like an important work almost

ready to be published.

The manuscript well describes the efforts and the work in well detailed & organized way.

Few comments to strengthen its impact are readable here below.

1. The text of the article has few grammar and typographic errors. The language of the manuscript better to be revised

2. It is uncommon to join Methods and finding together (Line 44)

3. It would be good to point to the burden of HIV with recent numbers (Line 132)

4. Inclusion and exclusion criteria need to be well defined and described

5. Although main finding well described, I wonder if other than descriptive conducted, need more elaboration

7. PLOS authors have the option to publish the peer review history of their article (what does this mean?). If published, this will include your full peer review and any attached files.

Reviewer #1: **Yes: **ISSAM J A ABU QEIS

Reviewer #2: No

---

## [Author Response · Author response to Decision Letter 1]

25 Oct 2021

Monday, October 25, 2021

Editors-in-Chief, PLOS ONE

Thank you for the review of our manuscript, “Leading causes of death and high mortality rates in an HIV endemic setting (Kisumu county, Kenya, 2019)” for publication in PLOS ONE. Thank you also for the positive feedback.

We have addressed the specific comments as follows: 

Reviewer #1: 

Comments: * statistical analysis was well prepared in this study

- it is interesting to have this conclusion, it gives a good matching between the background and its results.

- it's a well-written article by following a basic scientific English.

* I recommend to be published.

Response: Thank you for the recommendation and taking time to review our manuscript. 

Reviewer #2: 

Comments: Thank you for letting me revise such an interesting paper. It looks like an important work almost ready to be published.

The manuscript well describes the efforts and the work in well detailed & organized way.

Few comments to strengthen its impact are readable here below.

Comment #1. The text of the article has few grammar and typographic errors. The language of the manuscript better to be revised

Comment #2. It is uncommon to join Methods and finding together (Line 44)

Comment #3. It would be good to point to the burden of HIV with recent numbers (Line 132)

Comment #4. Inclusion and exclusion criteria need to be well defined and described

Comment #5. Although main finding well described, I wonder if other than descriptive conducted, need more elaboration

Responses: 

Comment #1: We have revised the text and corrected grammar and typographic errors. 

Comment #2: We have separated the headings 

Comment #3. We have separated the two points (population and HIV burden and provided the appropriate reference for each within the sentence)

Comment #4. We have separated the text under ‘study design and population’ and added a new section on ‘inclusion and exclusion criteria’. 

Comment #5. The reviewer notes that our analysis is largely descriptive in nature, and appears to be requesting additional analyses beyond those presented, though it is not clear from the comment which specific results the reviewer is requesting. We note that the findings presented are consistent with the study design, whose objectives were to ascertain UCOD, antecedent COD, and immediate COD for hospital-based deaths that occurred in two high-volume referral hospitals in Kisumu County, Kenya, and compare them to those notified to the civil registry, and finally to estimate mortality rates, hence we feel that the current scope and level of detail of the findings are appropriate given the objectives of the study.

We thank the reviewers again and hope that our paper will receive favorable consideration for publication.

Yours sincerely,

Anthony Waruru, for co-authors

---

## [Editor Report · Decision Letter 2]

26 Nov 2021

Leading causes of death and high mortality rates in an HIV endemic setting (Kisumu County, Kenya, 2019)

PONE-D-20-36117R2

Dear Dr. Anthony Waruru,

We’re pleased to inform you that your manuscript has been judged scientifically suitable for publication and will be formally accepted for publication once it meets all outstanding technical requirements.

Kind regards,

Sabeena Jalal, MBBS, MSc, MSc, SM

Academic Editor

PLOS ONE
---

## [Editor Report · Acceptance letter]

2 Dec 2021

PONE-D-20-36117R2 

Leading causes of death and high mortality rates in an HIV endemic setting (Kisumu county, Kenya, 2019) 

Dear Dr. Waruru:

I'm pleased to inform you that your manuscript has been deemed suitable for publication in PLOS ONE. Congratulations! Your manuscript is now with our production department. 

Kind regards, 

on behalf of

Dr. Sabeena Jalal 

Academic Editor

PLOS ONE